# A Piezoelectric Wave-Energy Converter Equipped with a Geared-Linkage-Based Frequency Up-Conversion Mechanism

**DOI:** 10.3390/s21010204

**Published:** 2020-12-30

**Authors:** Shao-En Chen, Ray-Yeng Yang, Guang-Kai Wu, Chia-Che Wu

**Affiliations:** 1Department of Mechanical Engineering, National Chung Hsing University, Taichung 40227, Taiwan; gear49761101@gmail.com (S.-E.C.); vrijheid235@gmail.com (G.-K.W.); 2Department of Hydraulic and Ocean Engineering, National Cheng-Kung University, Tainan 70101, Taiwan; ryyang@mail.ncku.edu.tw; 3Innovation and Development Center of Sustainable Agriculture (IDCSA), National Chung Hsing University, Taichung 40227, Taiwan

**Keywords:** wave-energy converter, piezoelectric power generator, frequency up-conversion mechanism, flexible piezoelectric composite film

## Abstract

In this paper, a piezoelectric wave-energy converter (PWEC), consisting of a buoy, a frequency up-conversion mechanism, and a piezoelectric power-generator component, is developed. The frequency up-conversion mechanism consists of a gear train and geared-linkage mechanism, which converted lower frequencies of wave motion into higher frequencies of mechanical motion. The slider had a six-period displacement compared to the wave motion and was used to excite the piezoelectric power-generation component. Therefore, the operating frequency of the piezoelectric power-generation component was six times the frequency of the wave motion. The developed, flexible piezoelectric composite films of the generator component were used to generate electrical voltage. The piezoelectric film was composed of a copper/nickel foil as the substrate, lead–zirconium–titanium (PZT) material as the piezoelectric layer, and silver material as an upper-electrode layer. The sol-gel process was used to fabricate the PZT layer. The developed PWEC was tested in the wave flume at the Tainan Hydraulics Laboratory, Taiwan (THL). The maximum height and the minimum period were set to 100 mm and 1 s, respectively. The maximum voltage of the measured value was 2.8 V. The root-mean-square (RMS) voltage was 824 mV, which was measured through connection to an external 495 kΩ resistive load. The average electric power was 1.37 μW.

## 1. Introduction

In recent decades, ocean waves have been considered as a potential candidate for renewable energy [1,2]. Wave energy has gradually attracted more and more attention from researchers [1,2]. According to the European Marine Energy Centre (MEMC), wave-energy convertors (WECs) can be classified into various categories. Three well-known types include wave-activated bodies, oscillating water columns (OWCs), and point absorbers. First, the wave-activated bodies use a power take-off (PTO) system driven by wave-induced motions of masses or large bodies to generate electricity. Numerous devices have been developed based on this concept; one well-known example is the Pelamis, developed by the Scottish company Pelamis Wave Power (formerly Ocean Power Delivery) in 2004 [3]. Henderson [4] would further study the Pelamis WEC to enhance control of the PTO system. Josset et al. [5], Ruellan et al. [6], and Cordonnier et al. [7] attempted to develop a SEAREV WEC based on this operational concept.

Secondly, OWCs consist of an air turbine and a chamber partially submerged below the water surface. The air in the chamber was compressed by the oscillating motion of the wave surface and flows through the air turbine to generate electric power. Falcao et al. [8] discussed the theoretical, numerical, and experimental modeling techniques associated with OWC converters. Lopez et al. [9] optimized the turbine-chamber coupling for the OWC via a 2D numerical model based on the RANS equations and the VOF surface-capturing scheme. Gomes et al. [10] proposed to improve the wave-energy extraction by the OWC by optimizing the dimensions of the floater and the tube under certain geometric constraints.

Finally, point-absorber WECs employ buoys excited by heavy wave motion to drive hydraulic pumps or linear generators to produce electric power. Zurkinden et al. [11] developed a nonlinear numerical model to analysis the most significant nonlinear effects of a point-absorber WEC; the dynamical properties of the semi-submerged hemisphere buoys, oscillated by waves, were investigated. Bozzi et al. [12] presented a numerical model of the coupled buoy-generator system to simulate the behavior of the WEC under different wave heights and periods. This numerical model based on linear-potential-wave theory simulated the influence of the hydrostatic forces upon a point-absorber WEC, and also considered the radiation impedance and excitation force. Nevertheless, those WECs had heavy, bulky generators such as linear generators, rotary generators, or hydraulic pumps.

Piezoelectric materials, which have the advantages of a small size and large energy density, have been used in applications such as wind and vibration energy [13,14,15,16]. Kan et al. [17] proposed a piezoelectric windmill that could harvest wind energy at low speeds and over a wide speed range. The operational frequency of the piezoelectric component was increased by excitation with a rotating magnet. A prototype of this piezoelectric windmill was fabricated and tested to prove the analytical result. Zhao et al. [18] optimized the performance of the galloping piezoelectric energy harvester using an effective analytical mode to incorporate both electromechanical coupling and the aerodynamic force. Dai et al. [19] investigated a piezoelectric energy harvester consisting of a multilayered piezoelectric cantilever beam, which was applied to energy harvesting from base excitations and vortex-induced vibrations. Mutsuta et al. [20] proposed a wave-energy harvester using painted flexible piezoelectric device (FPED) that had an elastic material deformed by the wave and a piezoelectric paint to generate electric power. The FPED was tested in various wave conditions to prove its function and the experimental results had good agreement with the proposed theoretical model. Fan et al. [21] proposed a nonlinear harvester capable of collecting energy from various vibration directions. The low-frequency vibration in the environment was converted into high-frequency motion by improving performance through the action of magnetic coupling. In this paper, we developed a WEC with a piezoelectric generator. The size of the WEC equipped with a piezoelectric generator might be reduced, potentially enhancing the electrical output by parallel or series circuits.

However, the use of piezoelectric materials in wave environments requires the challenge of lower driving frequencies that are far from the natural frequency of the piezoelectric component to be overcome. Li et al. [22] discussed the application and operation of piezoelectric energy harvesters in low-frequency environments (0–100 Hz). There are three methods for applying piezoelectric materials to low-frequency environments, including frequency tuning, broadband response, and frequency up-conversion techniques. Fan et al. [23] designed and developed a piezoelectric energy harvester using a beam-roller configuration to convert low-frequency sway and vibration into high-frequency vibration of the piezoelectric beam. Another approach was to use the flexible substrate to obtain larger deformation of the piezoelectric material to enhance the piezoelectric effect. Lin et al. [24] proposed a wave-energy harvester with a mechanical impact-driven frequency up-converted device. A mathematical model of this device was established, and compared with experimental results. The results showed that the varying frequency of the waves could be converted into a higher preset frequency of the vibration of the beams. Renzi [25] derived a fully coupled model for investigating the hydroelectromechanical-coupled dynamics of a piezoelectric wave-energy converter. The relationship between the plate motion and power extraction was also determined by the mathematical model. Yang et al. [26] developed a prototypical vibration-energy-harvesting system using a large-fiber composite material distinct from lead zirconate titanate; this new material has the property of being flexible under large deformation. Orrego et al. [27] developed a wind-energy harvester to collect energy through self-sustained oscillations of a flexible piezoelectric membrane. They evaluated the flapping behavior and resultant energy output by studying the influence of the geometrical parameters. In a wave environment, increasing the drive frequency through the design of a frequency up-conversion mechanism and increasing the deformation of the piezoelectric material can greatly improve the material’s power-generation efficiency.

Existing WECs utilize linear generators, rotary generators driven by mechanical linear-to-rotary converters, or hydraulic pumps. However, these generators are bulky and heavy. In this study, a novel WEC using a piezoelectric power-generation component, including a flexible piezoelectric composite film (piezoelectric film) is developed. Compared with the generators mentioned above, a piezoelectric film offers a small size and simple structure. To achieve larger electrical power, higher operating frequencies and larger deformation in the deflection range of the piezoelectric film are preferable. A frequency up-conversion mechanism based on a geared-linkage mechanism is developed to convert low-frequency wave motion into higher-frequency mechanical motion. Mechanical deformation with limited amplitude is used to drive the piezoelectric generator. In this work, the piezoelectric performance of the film is tested via scanning electron microscopy, X-ray diffraction, and capacitance testing. The kinematic performance of the frequency up-conversion mechanism is analyzed by computer and tested in a wave flume. The analytical and experimental results are compared. Finally, the electrical output of the PWEC is measured via an actual wave test in the wave flume. The output voltage, RMS voltage, and average electric power are discussed.

## 2. Conceptual Design

### 2.1. The Components of the Piezoelectric WEC

In this study, a PWEC consisting of a buoy, a frequency up-conversion mechanism, and a piezoelectric power-generation component is developed, as shown in Figure 1. Figure 1a shows the heaving buoy on the ocean surface excited by the wave motion, driving the rack for heave motion. Wave energy is absorbed by the buoy and converted into mechanical energy to drive the frequency up-conversion mechanism. We choose Styrofoam as a material for the buoy because it has a small density that easily floats on the surface and responds quickly to changes in the water level. The geometry of the buoy is designed as a sphere with a diameter of 40 cm. The rack was obtained from APEX CYNAMICS, Inc. (Taichung, Taiwan), with lengths and pitch of 100 mm and 1 mm, respectively.

The gear-linkage-based frequency up-conversion mechanism is shown in Figure 1b; it includes a rack, a gear train, and a geared-linkage mechanism. The rack with repetitive linear motion drove the gear train for rotational motion. The gear train included a gear and a pinion with a specific gear ratio to adjust the tangential speed between the gear and pinion. The gear and pinion are mounted on the same rotating shaft and the pinion is driven directly by the rack. The gear and pinion have the same angular velocity. The gear and pinion were obtained from APEX CYNAMICS, Inc. The gear and pinion have 75 teeth and 25 teeth, respectively. The module of the gear and pinion is 1.

The geared-linkage mechanism was then driven by the gear train, which converted the rotational motion of the pinion into the repetitive motion of the slider. The geared-linkage mechanism included a pinion, a disk, a linkage, and a slider. The number of teeth and the module of this pinion are 25 and 1, respectively. Because the gear ratio between the pinion of the geared-linkage mechanism and the gear of the gear train is 3, the pinion rotates three times for each rotation of the gear. A disk was mounted on the pinion as a crank link for the slider-crank. The slider has a higher speed and frequency than the wave motion and was used to move the fixture of the piezoelectric power-generation component, as shown in Figure 1c. The slider-crank and the disk of the geared-linkage mechanism are made of transparent acrylic material. This offers the advantages of high mechanical strength, light weight, ease of machining, and low cost. A typical slider-crank consists of a crank link, a coupler link, and a slider. When the crank link is driven through a full 360° rotation, the slider moves forward and backward for one cycle. The defined displacement range of the traveling slider is determined by the length of the crank link. In this work, this length (radius of the disk) is 15 mm and the length of the coupler link is 35 mm. The range of the slider is 30 mm.

In this study, the gear ratio of the gear train was 3, meaning that the pinion rotates 3 times for each rotation of the gear. The displacement range of the linear repetitive motion was 30 mm. When a wave had traveled after one complete cycle, the rack-driven gear completed two cycles of rotary motion, one clockwise and one counterclockwise. During the wave’s period, the gear rotated for two cycles, and the pinion rotated for six. Finally the slider has a six-period displacement. Therefore, the operational frequency of the piezoelectric power-generation component driven by the slider is six times the wave-motion frequency.

### 2.2. Operation of the Piezoelectric Power-Generation Component

In this work, the piezoelectric power-generation component included a fixed part and a moving part, as shown in Figure 2a. The fixed part was used to clamp the piezoelectric film. The moving part was driven by the slider to deform the piezoelectric film. A cross-sectional view of the A–A line in Figure 2a is shown in Figure 2b. The piezoelectric film was undeformed when the slider was located at the equilibrium position. When the slider moved, the moving part of the fixture would deform the piezoelectric film, as shown in Figure 2c,d. When the slider moves upward, the piezoelectric film bends with a positive moment. On the other hand, a negative bending moment was exerted when the slider moved downward. The piezoelectric film generates electrical power through the piezoelectric effect. The fixture was made of acrylic because this material offers high mechanical strength, low cost, ease of machining, and transparency.

### 2.3. Flexible Piezoelectric Composite Film

The developed, flexible piezoelectric composite films (piezoelectric films) were used to generate electrical voltages at a larger deformation and a lower driving frequency. The film was clamped to the fixture, and the four-plate structures were deformed by the moving part. The longest dimension of the film was 130 mm; the length and width of the plate structures were 26 mm and 78 mm, respectively. The cross section of the piezoelectric film consisted of copper (Cu)/nickel (Ni) foil, a lead–zirconium–titanium (PZT) piezoelectric-material layer, and a silver upper-electrode layer. The substrate made of Cu/Ni foil was obtained from APEX CYNAMICS, Inc., and offered good electrical conductivity and flexibility. The PZT layer was made in the lab. The top electrode was silver. The thicknesses of the copper substrate, nickel substrate, PZT, and silver upper-electrode layers were 100 μm, 10 μm, 158 μm, and 10 μm, respectively. The optimized thickness of the PZT layer was determined by calculating the neutral axis. To maximize the voltage output of the piezoelectric material, the position of the neutral axis was located in the substrate layer. The thickness and mechanical properties of each material layer of the film are shown in Table 1. The piezoelectric constant of the piezoelectric layer is also shown in Table 1.

## 3. Design, Analysis, and Testing of the PWEC

### 3.1. Kinematics Analysis of the Frequency Up-Conversion Mechanism

There are two goals for the frequency up-conversion mechanism designed in this study. First, the low-frequency motion of the wave should be converted into a higher-frequency mechanical vibration to drive the piezoelectric film, through design of the gear ratio. Second, the amplitude of the mechanical vibration should be limited, even when driven by large-amplitude waves, to prevent damage (destruction, cracking, breakage, and failure) of the piezoelectric film. A theoretical mathematical model and a software-based numerical model are developed and studied.

To simplify the mathematical model, the weight and the friction were neglected. A schematic diagram of the geared-linkage mechanism is shown in Figure 3. The points O, A, and B are the revolution joints that connect the disk to a crank link, a coupler link, and a slider. The displacements $\vec{R_{1}}$, $\vec{R_{2}},$ and $\vec{R_{3}}$ are the vector functions from O to B, from O to A, and from B to A, respectively. The norms of the vectors $\vec{R_{1}}$, $\vec{R_{2}},$ and $\vec{R_{3}}$ are *a*, *b*, and *c*. $\theta_{1}$ and $\theta_{2}$ are angles from the *x*-axis to vector $\vec{R_{2}}$ and from the *x*-axis to vector $\vec{R_{3}}$, respectively. Since the sum of vectors $\vec{R_{1}}$, $\vec{R_{2}},$ and $\vec{R_{3}}$ is zero, the closure equations are shown as follows:(1)$$b\left(\cos\theta_{1}+j\sin\theta_{1}\right)-c\left(\cos\theta_{2}+j\sin\theta_{2}\right)-a=0;$$
(2)$$a=b\cos\theta_{1}-c\cos\theta_{2};$$
(3)$$\theta_{2}=\sin^{-1}\left(\frac{b\sin\theta_{1}}{c}\right).$$

The schematic diagram of the frequency up-conversion mechanism is shown in Figure 3. This mechanism includes a rack, a gear train, and a geared-linkage mechanism. The rack was excited by wave motion and drove the gear train for rotational motion. We assumed a single sinusoidal wave motion. The displacement and velocity of the rack can be written as
(4)$$x_{R}\left(t\right)=A\sin\left(\omega t+\varphi \right),$$
(5)$$V_{R}\left(t\right)=A\omega \cos\left(\omega t+\varphi \right),$$
where A is the maximum amplitude of the buoy, ω is the angular frequency, t is the time, and φ is the phase angle. The gear train included a gear of radius $r_{G}$ and a pinion of radius $\;r_{PT}$. The pinion of the gear train was driven by the rack. The tangential velocity of the pinion$,\;\;V_{PT}\left(t\right),$ was equal to that of the rack $\;V_{R}\left(t\right)$. The angular velocity of the pinion $\;\Omega_{PT}\left(t\right)$ can be expressed as follows:(6)$$\Omega_{PT}\left(t\right)=\frac{V_{PT}\left(t\right)}{r_{PT}}=\frac{A\omega \cos\left(\omega t+\varphi \right)}{r_{PT}}.$$

The pinion and the gear of the gear train were mounted coaxially. The angular velocity of the gear, $\Omega_{G}\left(t\right),$ was equal to that of the pinion$,\;\Omega_{PT}\left(t\right)$. Therefore, the tangential velocity of the gear $\;V_{G}\left(t\right)$ can be written as
(7)$$V_{G}\left(t\right)=\Omega_{G}\left(t\right)r_{G}=\frac{r_{G}}{r_{PG}}A\omega \cos\left(\omega t+\varphi \right).$$

The geared-linkage mechanism included a pinion, a disk, a coupler link, and a slider. The pinion was driven by the gear of the gear train with the same tangential velocity. The angular velocity, $\Omega_{PL}\left(t\right),$ of the pinion of the geared-linkage mechanism can be written as
(8)$$\Omega_{PL}\left(t\right)=\frac{r_{G}}{r_{PG}r_{PL}}A\omega \cos\left(\omega t+\varphi \right).$$

$\theta_{1}\left(t\right)$ can be obtained by integrating the angular velocity of the pinion of the geared-linkage mechanism:(9)$$\theta_{1}\left(t\right)=\int^{}\Omega_{PL}\left(t\right)dt=\frac{r_{G}}{r_{PG}r_{PL}}A\sin\left(\omega t+\varphi \right).$$

The displacement of the slider can be obtained by (2), (3), and (9):(10)$$a=b\cos\left(\frac{r_{G}}{r_{PG}r_{PL}}A\sin\left(\omega t+\varphi \right)\right)-c\cos\left(\sin^{-1}\left(\frac{b\sin\left(\frac{r_{G}}{r_{PG}r_{PL}}A\sin\left(\omega t+\varphi \right)\right)}{c}\right)\right).$$

In this work, $r_{G}$ is 37.5 mm, $r_{PT}$ is 12.5 mm, $r_{PL}$ is 12.5 mm, b is 15 mm, and c is 35 mm. When the wave frequency was 1 Hz, the frequency of the slider was 6 Hz. Moreover, the amplitude of the slider would be smaller than 30 mm. Therefore, the maximum deflection of the piezoelectric film was 30 mm to avoid fracturing the film.

### 3.2. The Flexible Piezoelectric Composite Film

The flexible piezoelectric composite films were fabricated for low frequency and large deformation. To achieve lower natural frequency and larger bending deformation, copper–nickel foil was used as the substrate instead of a silicon wafer. The low Young’s modulus (E = 120 GPa) and large tensile strength (σ= 267 MPa) of the copper–nickel foil were preferred. PZT with excellent piezoelectric properties was used in this work. In our previous work, flexible PZT composite films were fabricated on the copper/polyimide flexible substrate by the sol-gel process, which is a promising fabrication method [28].

#### 3.2.1. Preparation of the PZT Slurry

The fabrication process of the PZT slurry obeyed the following steps:(a)1.6 g of PVB powder was added into 44 mL of a 99.9% alcohol solution via vigorous stirring at 600 rpm for 1 h.(b)80 g of PZT NPs were added to the PVB–alcohol solution and stirred vigorously at 1000 rpm for 1 h.(c)The solution was subjected to ultrasonic mixing in an oscillating machine for 1 h.(d)The solution was vigorously stirred again at a rate of 1000 rpm for 30 min to complete the slurry preparation.

#### 3.2.2. Coating and Sintering Process of the PZT Film

The coating and sintering process obeyed the following steps.

(a)The copper–nickel foil was patterned in the desired shape.(b)This flexible substrate (Cu/Ni) was rinsed and cleaned with ethanol and DI water via ultra-sonication for 15 min and dried by nitrogen.(c)The flexible substrate (Cu/Ni) was baked on a hotplate at 100 °C for 5 min.(d)The spin coater was used to coat the slurry on the substrate at 350 rpm for 20 s.(e)The low-temperature sintering method was used via heating the piezoelectric film on a hotplate at 100 °C for 10 min.(f)The sample was cooled down to room temperature.(g)The sample was cleaned with a nitrogen gun.(h)Repeat steps (d)–(g) to reach the required thickness.

After deposition of the PZT on a flexible substrate, a silver paste was screen printed on the piezoelectric film as the top electrode. The silver top electrode was formed on the flexible composite film at 130 °C for 15 min. After the fabrication process was completed, an electric field of 3.1 kV/cm was applied to the piezoelectric film at 90 °C for 30 min to polarize the piezoelectric film.

### 3.3. Experimental Setup

The experiment was performed in the wave flume at the Tainan Hydraulics Laboratory, National Chung Kung University, Taiwan (THL, NCKU). The total length, width, and depth of the wave flume were 27 m, 19 m, and 1 m, respectively. This flume was also equipped with an active wave-absorption function and an electro-serve motor, feedback-control system wave generator, producing waves up to 0.3 m high at a period of 1.0 s to 4.0 s. The wave generator with a plate structure of 7 m in width and 1.2 m in height was applied for wave generation. Two target wave heights (100 mm and 75 mm) and two periods (1 s and 1.5 s) were used. The 4 cases are shown in Table 2.

A schematic view of the experimental setup is shown in Figure 4. A capacitive wave-height meter was equipped near the water’s surface to measure the change in wave height. The developed PWEC was placed in the wave flume and generated electrical power when a wave passed through. LEDs A and B and a video recorder were used to determine the displacement of the moving part. LED A was placed on the buoy and LED B was placed on the slider. A video recorder was set at one side of the wave flume. The displacements of the buoy and the slider were obtained from analysis of the video image. An oscilloscope was used to measure the voltage output generated by the PWEC. The RMS voltage was calculated. The average electric power was calculated by the following equation:(11)$$P_{ave}=\frac{V_{rms}{}^{2}}{R_{L}},$$
where $P_{ave}$ is the average electric power, $V_{rms}$ is the RMS voltage, and $R_{L}$ is the resistive load. A bridge-rectifier circuit was used to rectify the electrical current for further power storage.

## 4. Results and Discussions

### 4.1. Characterization of the PZT Film

The developed flexible piezoelectric composite film is shown in Figure 5a. A scanning electron microscope was used to monitor the morphology of the film, as shown in Figure 5b. The PZT film was extremely dense and the average particle size was less than 1 μm. An X-ray diffractometer was used to examine the film’s crystalline structure. Patterns were recorded at a rate of 4° min^−1^ in the 2θ range of 20° to 70°. The X-ray-diffraction (XRD) results of the PZT film are shown in Figure 5c. A significant peak was observed at 31.00°, indicating that the piezoelectric film had a perovskite crystal. A capacitance test was used to characterize the electrical properties of the PZT film by measuring the capacitance and dissipation factor. In the capacitance test, a LCR meter (HP LCR 4284A) drove the PZT film with a sinusoidal voltage from 40 Hz to 3 kHz. The LCR meter recorded the current and plotted the capacitance and dielectric loss as a function of driving the voltage simultaneously. The result of the capacitance test is shown in Figure 5d. The capacitance of the PZT film was 39.77 nF at 1 kHz. The dissipation loss was 0.0175 at 1 kHz. Capacitance is the ability to store electrical energy, and dissipation is the measure of the energy-loss rate in a mode of the electrical dissipative system.

### 4.2. Comparison with Wave Motion

#### 4.2.1. Movement between the Buoy and the Wave

In this work, the developed PWEC was tested in the wave flume at the Tainan Hydraulics Laboratory, National Cheng Kung University, Taiwan (THL, NCKU). The wave height generated was measured by a capacitive wave-height meter. The displacements of the buoy and the slider of the geared-linkage mechanism were recorded by LED A, LED B, and a video recorder. Their displacement results were obtained from image analysis. Four results (Cases 1–4) are shown in Figure 6. The amplitudes of the measured waves were 67 ± 0.841 mm, 52 ± 0.948 mm, 67 ± 0.884 mm, and 51 ± 1.166 mm, respectively. When the wave generator was equipped with an electro-serve motor, feedback-control system, the actual amplitude was 15 mm larger than the control wave in Case 1. This may be due to the evolution of waves over times, the time delay of the control system, and the geometry of the wave flume.

The amplitudes of the buoy displacement were 34 ± 1.052 mm, 27 ± 1.030 mm, 37 ± 1.068 mm, and 29 ± 1.219 mm for Cases 1–4, respectively. The amplitude of buoy displacement was around 30% less than the actual wave amplitude. This difference between the wave amplitude and the buoy displacement was because of the response amplitude operator (RAO) generally being less than 1.0, about 0.7. The distributed pressure on the surface of the buoy results in a lower force, thus leading to a lower displacement than the wave. In future research, the applied buoy should be analyzed by hydrodynamics to improve the RAO.

#### 4.2.2. The Performance of the Frequency Up-Conversion Mechanism

In this study, the kinematic properties of the developed frequency up-conversion mechanism were tested. The experimental results were compared to the analytical ones. The displacement of the moving part of the fixture connected to the slider was measured by video. Experimental results for Cases 1–4 are shown in Figure 7 and were similar to the analytical results. Even the trend was similar; there was still a reasonable difference between the experimental and analytical results. This might be attributable to manufacture tolerance, assembly offset, and friction.

The frequency spectra of the buoy and slider motion obtained by fast Fourier transformation (FFT) of Case 1 data is shown in Figure 8. The frequency of the buoy motion was 1.17 Hz, which is higher than that of the control wave (1 Hz). This may be the result of error in the control system of the wave generator. Another possibility might be the small size of the data sample. The resolution of the frequency spectrum was limited by the sampling data, and only 0.97 Hz and 1.17 Hz were calculated by FFT. The actual resonance frequency might be located between 0.97 Hz and 1.17 Hz. The main frequency of the slider motion was 6.44 Hz, which was six times larger than that of the buoy. This result proves that the frequency up-conversion mechanism converts low-frequency wave motion into mechanical motion with a frequency six times greater.

### 4.3. Electrical Performance of the PWEC

In this study, the electrical performance of the developed PWEC, including the voltage and current outputs, were measured by an oscilloscope. The output voltages of the PWEC for Cases 1–4 are shown in Figure 9a–d, respectively. The maximum voltage in each case is summarized in Table 3. The maximum voltages were 2.8 V, 2.28 V, 2.24 V, and 2.02 V for Cases 1–4, respectively. The RMS voltage equivalent to the DC voltage was measured through connecting to an external 495 kΩ resistive load. Table 3 shows the RMS voltage in each case. These voltages were 824 mV, 595 mV, 630 mV, and 506 mV for Cases 1–4, respectively. The average electric powers in Table 3 were calculated by Equation (11) and had values of 1.37 μW, 0.71 μW, 0.8 μW, and 0.52 μW, respectively. It can be seen that the PWEC functioned well and outputted electrical energy. Among the four cases, Case 1 had the largest RMS voltage (824 mV) and average electric power (1.37 μW). Recall in Figure 6a, Case 1 had a larger and more frequent displacement. The wave energy was larger and the output voltage was higher.

This study offered three advantages. First, the developed frequency up-conversion mechanism converted low-frequency wave motion into higher-frequency mechanical motion to improve the electrical output of the PWEC. Second, the range of the slider limits the deformation of the piezoelectric film, allowing it to avoid fracture for any wave height. Finally, the flexible piezoelectric composite film had good flexibility that improved the electrical output of the PWEC.

## 5. Conclusions

A PWEC consisting of a buoy, a frequency up-conversion mechanism, and a piezoelectric power-generator component was developed. The operational frequency of the piezoelectric power-generation component driven by the frequency up-conversion mechanism was six times that of the wave motion. The developed, flexible piezoelectric composite films of the generator component were used to produce electrical voltage under a low driving frequency and larger deformation. The deformation range was 30 mm, limited by the range of the slider of the geared-linkage mechanism. This limitation allowed fracturing of the flexible piezoelectric composite film to be avoided.

The developed PWEC was tested in the wave flume at the Tainan Hydraulics Laboratory, National Cheng Kung University, Taiwan (THL, NCKU). Two target wave heights (100 mm and 75 mm) and two periods (1 sec and 1.5 sec) were used. The amplitudes of the measured waves were 67 ± 0.841 mm, 52 ± 0.948 mm, 67 ± 0.884 mm, and 51 ± 1.166 mm for Cases 1–4, respectively. The amplitudes of the buoy displacement were ≈34 ± 1.052 mm, ≈27 ± 1.030 mm, ≈37 ± 1.068 mm, and ≈29 ± 1.219 mm for Cases 1–4, respectively. The frequencies of the buoy and slider motion obtained by FFT in Case 1 were 1.17 Hz and 6.44 Hz, respectively. These results led to the conclusion that low-frequency wave motion was converted into mechanical motion with a six-times-higher frequency. The output voltages of the developed PWEC for Cases 1–4 were measured by an oscilloscope. The maximum voltages were 2.8 V, 2.28 V, 2.24 V, and 2.02 V for Cases 1–4, respectively. The RMS voltage was measured through connection to an external 495 kΩ resistive load. The RMS voltages were 824 mV, 595 mV, 630 mV, and 506 mV for Cases 1–4, respectively. The average electric powers were 1.37 μW, 0.71 μW, 0.8 μW, and 0.52 μW, respectively. Case 1 had the largest RMS voltage (824 mV) and average electric power (1.37 μW). These results indicated that the PWEC functioned well and outputted electrical energy. The PWEC will be tested under random waves in the future. We expect that the PWEC will be able to generate electricity when the wave amplitude is greater than 15 mm, which can drive the slider to have a 30 mm displacement. The period of output voltage will depend on the frequency of the waves.

## Figures and Tables

**Figure 1 sensors-21-00204-f001:**
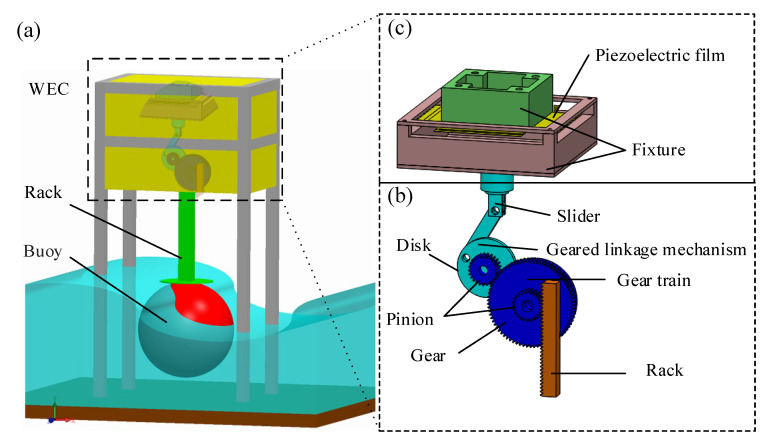
Schematic diagrams of the piezoelectric wave-energy converter. (**a**) Piezoelectric wave-energy converter; (**b**) frequency up-conversion mechanism; (**c**) piezoelectric power-generation component.

**Figure 2 sensors-21-00204-f002:**
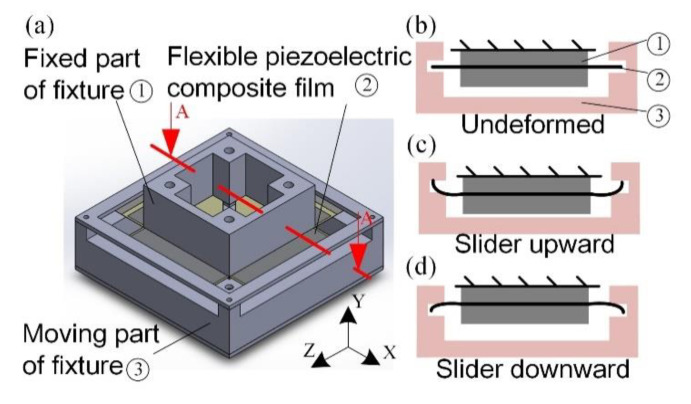
Schematic diagrams of the piezoelectric power-generation component. (**a**) Piezoelectric-power-generation component; (**b**) slider located at the equilibrium point; (**c**) slider upward; (**d**) slider downward.

**Figure 3 sensors-21-00204-f003:**
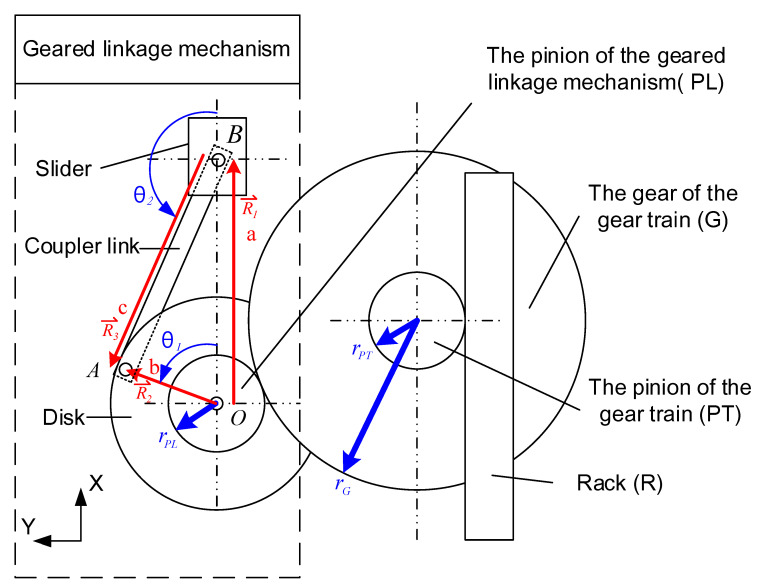
Schematic diagram of the frequency up-conversion mechanism.

**Figure 4 sensors-21-00204-f004:**
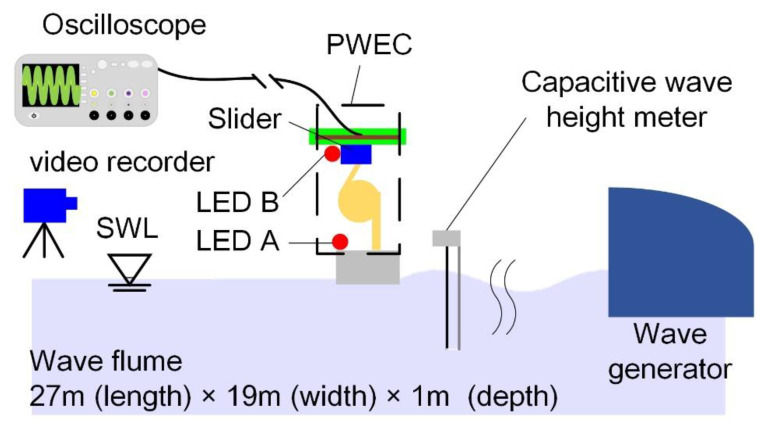
Schematic view of the experimental setup.

**Figure 5 sensors-21-00204-f005:**
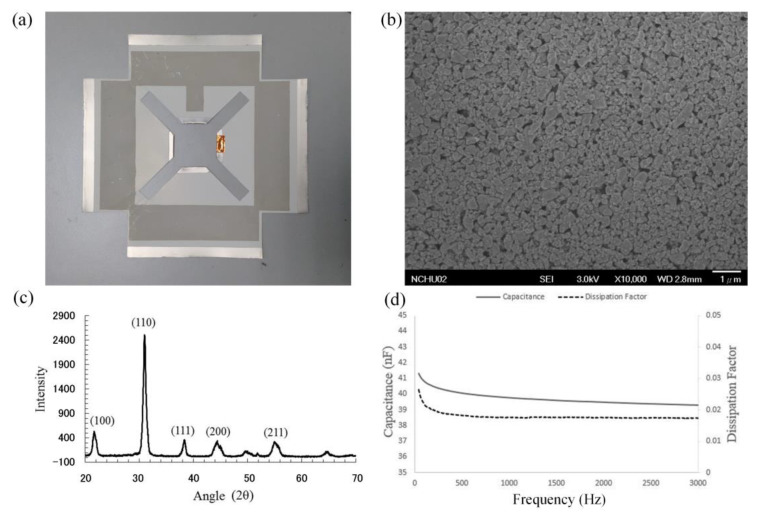
The result of the flexible piezoelectric composite film. (**a**) The four-plate structures of piezoelectric film; (**b**) SEM image; (**c**) X-ray diffraction (XRD) results; (**d**) capacitance test.

**Figure 6 sensors-21-00204-f006:**
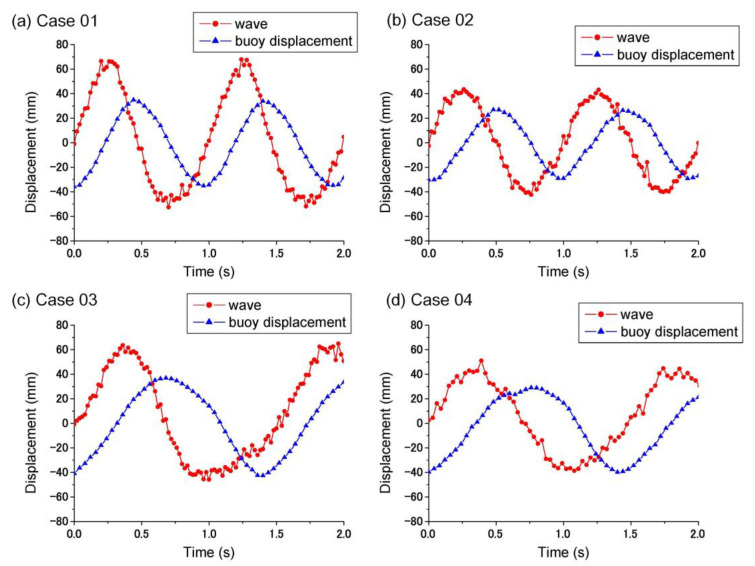
Wave height and buoy displacement.

**Figure 7 sensors-21-00204-f007:**
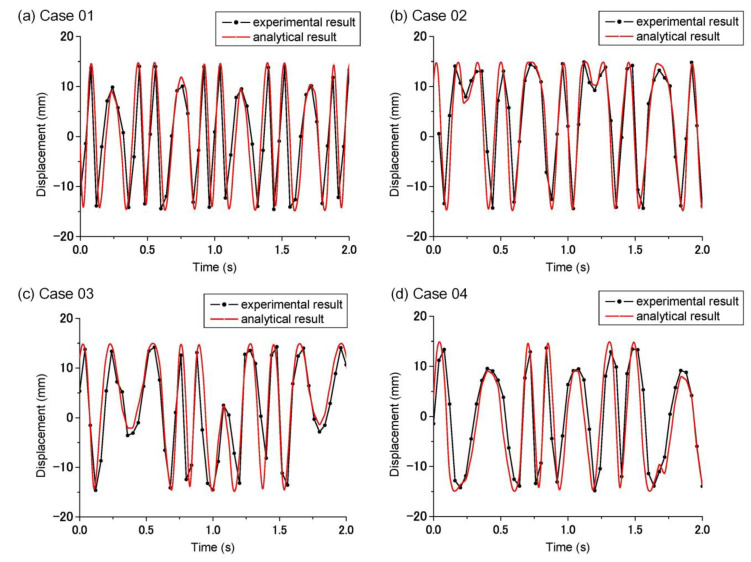
The displacement result of the slider.

**Figure 8 sensors-21-00204-f008:**
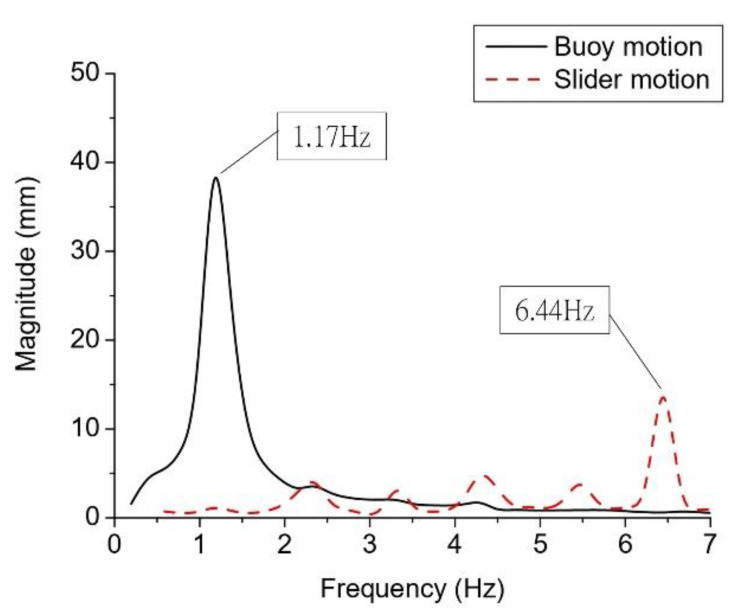
Spectra of the slider and buoy motion as a function of wave frequency in Case 1.

**Figure 9 sensors-21-00204-f009:**
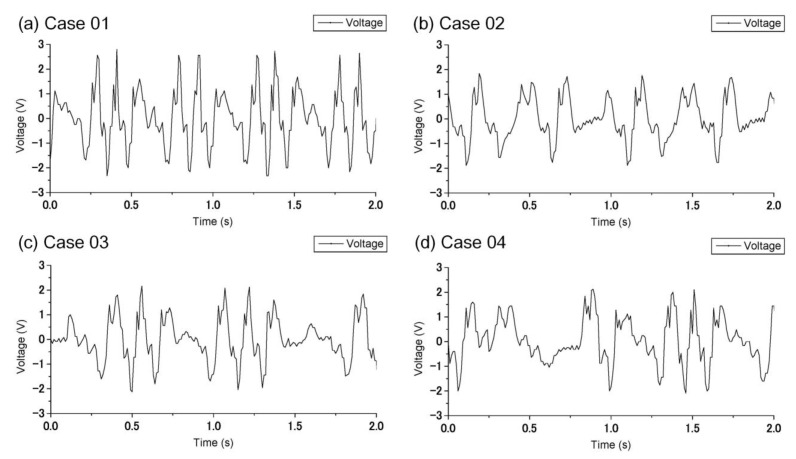
The measured voltage output of the PWEC in each case.

**Table 1 sensors-21-00204-t001:** The thickness and mechanical properties of each material layer of the film.

Material	Copper (Cu)	Nickel (Ni)	PZT	Silver (Ag)
Thickness (μm)	100	10	158	10
Young’s modules (GPa)	128	200	67	83
Density (kg/m^3^)	8960	8908	7800	10,490
Poisson’s ratio	0.34	0.31	0.39	0.37
Piezoelectric constant (pm/V)	N/a	N/a	−210	N/a

**Table 2 sensors-21-00204-t002:** Wave-period and wave-height-parameter table generated by the wave machine.

Wave Height (mm)	100	75
Period (1 s)	Case 01	Case 02
Period (1.5 s)	Case 03	Case 04

**Table 3 sensors-21-00204-t003:** Electrical output in each case.

	Case 01	Case 02	Case 03	Case 04
Maximum voltage (V)	2.8	2.28	2.24	2.02
RMS Voltage (mV)	824	595	630	506
Average electric power (μW)	1.37	0.71	0.8	0.52

## Data Availability

Not applicable.

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
