# Peer review of "A Piezoelectric Wave-Energy Converter Equipped with a Geared-Linkage-Based Frequency Up-Conversion Mechanism"

_sensors, 2020, doi:10.3390/s21010204_

Round 1

Reviewer 1 Report

Dear authors,

thank you for possibility to review your paper related to the PWEC in practical solution. The paper is well written, I have few comments/questions:

1) Can you estimate the power looses or shifts in the frequency-up-conversion mechanism with a rack, a gear train, and a geared-linkage mechanism?

2) You are using Copper, Nickel, PZT ang Silver as basic material for the piezoelectric-power-generation component. Some of these material will not persist the aggressive environment like the ocean apparently is. Have you tried long-term resistance?

3) Figures quality (mainly Fig. 6-8) is very low and need improvement.

4) There is few repeating typo errors (spaces between number and units, if the number is not exact, or is measured, there should be ≈ symbol, otherwise there should be (mean ± standard deviation) value.

For this reasons, I suggest to accept paper with minor revision.

Reviewer 2 Report

This paper reports a piezoelectric wave-energy converter (PWEC) to harvest wave energy. The converter is rationally designed consisting of a buoy, a frequency up-conversion mechanism, and a piezoelectric-power-generator component. Lead-zirconium-titanium (PZT) film is used as piezoelectric layer to convert mechanical energy to electricity. Before recommending its publication on Sensors, some issues should be considered. Please carefully address the following concerns. 1. The PZT layer is spin coated on Cu/Ni substrate with thickness of 158 um. How do the authors know the thickness of PZT layer? SEM image of the cross section of Cu/Ni/PZT/Ag film is suggested to be offered. 2. Although the authors claim that the range of the slider limits the deformation of piezoelectric film, the reviewer is still in concern. Cross sectional SEM images of Cu/Ni/PZT/Ag film after a certain cycles of working might help us to check if the film crash or not. 3. Figure 6 and 7are a little bit fuzzy, high definition graph should be offered.

Round 2

Reviewer 2 Report

The authors have addressed my concerns, it can be accepted in the present format.